# Low-coherence subspace projection: enhance the learning capacity of orthogonal projection methods on long task sequences

## Abstract

Gradient Orthogonal Projection (GOP) is an efficient strategy in continual learning to mitigate catastrophic forgetting. Despite its success so far, GOP-based methods often suffer from the learning capacity degradation problem with an increasing number of tasks. To address this problem, we propose a novel and plug-and-play method to learn new tasks in low-coherence subspaces rather than orthogonal subspaces. Specifically, we construct a unified cost function with the DNN parameters lying on the Oblique manifold. A corresponding gradient descent algorithm is developed to jointly minimize the cost function that involves both inter-task and intra-task coherence. We then provide a theoretical analysis to show the advantages of the proposed in the stability and plasticity. Experimental results show that the proposed method has prominent advantages in maintaining the learning capacity, when the number of tasks increases, especially on a large number of tasks, compared with baselines.

## 1 Introduction

Although **D**eep **N**eural **N**etworks (DNNs) have achieved promising performance in many tasks, their applications are limited for continual learning, suffering from catastrophic forgetting French (1999). When tasks are to be learned sequentially, catastrophic forgetting refers to the phenomenon of new knowledge interfering with old knowledge. Research in continual learning, also known as incremental learning Aljundi et al. (2018a); Chaudhry et al. (2018a); Chen & Liu (2018); Aljundi et al. (2017), and sequential learning Aljundi et al. (2018b); McCloskey & Cohen (1989), aims to find effective algorithms that enable DNNs to simultaneously achieve plasticity and stability, i.e., to achieve both high learning capacity and high memory capacity.

Various methods have been proposed to avoid or mitigate the catastrophic forgetting De Lange et al. (2019), either by replaying training samples Rolnick et al. (2019); Ayub & Wagner (2020); Saha et al. (2021), or reducing mutual interference of model parameters, features or model architectures between different tasks Zenke et al. (2017); Mallya & Lazebnik (2018); Wang et al. (2021). Among these methods, **G**radient **O**rthogonal **P**rojection (GOP) Chaudhry et al. (2020); Zeng et al. (2019); Farajtabar et al. (2020); Li et al. (2021) is an efficient continual learning strategy that advocates projecting gradients with the orthogonal projector to prevent the knowledge interference between tasks. GOP-based methods have achieved encouraging results in mitigating catastrophic forgetting. However, from Fig. 1, we observe that these methods suffer from the learning capacity degradation problem. Namely, their learning capacity is gradually degraded as the number of tasks increases and eventually becomes unlearnable. Specifically, when learning multiple tasks, e.g., more than 30 tasks in Fig. 1, their performance on new tasks dramatically decreases. These results suggest that the GOP-based methods focus on the stability and somewhat ignore the plasticity. Ignoring the plasticity may limit the task-learning capacity of models, i.e., The performance of the model on a new task when learning multiple tasks consecutively.

To address this issue, we propose a novel projection-based method, called **L**ow-**c**oherence **S**ubspace **P**rojection (LcSP), which learns new tasks in low-coherence subspaces rather than orthogonal subspaces. Specifically, LcSP utilizes low-coherence projectors at each layer to project both features and gradients into subspaces with low coherence. To achieve this, we construct a unified cost

function to find projectors and develop a gradient descent algorithm on the Oblique manifold to jointly minimize inter-task coherence and intra-task coherence among the projectors. Minimizing the inter-task coherence can reduce the mutual interference between tasks while minimizing the intra-task coherence can enhance the model's expressive power. Restricting projectors on the Oblique manifold can avoid the scale ambiguity Aharon et al. (2006); Wei et al. (2017), i.e., preventing the parameters of the projector from being extremely large or extremely small. Moreover, the algorithm we propose for constructing low-coherence projectors is a plug-and-play module. By reusing this module, LcSP can be easily extended to most GOP methods. For example, based on LcSP, we provide the algorithm pseudo-code for continual learning with GPM Saha et al. (2021), which can be used in both task-incremental and class-incremental settings, in the Appendix A.2.

## 2  RELATED WORK

In this section, we briefly review some existing works of continual learning and the GOP-based methods.

**Replay-based Strategy.** The basic idea of replay-based approaches is to use limited memory to store small amounts of data (e.g., raw samples) from previous tasks, called episodic memory, and to replay them when training a new task. Some of the existing works focused on selecting a subset of raw samples from the previous tasks Rolnick et al. (2019); Isele & Cosgun (2018); Chaudhry et al. (2019); Zhang et al. (2020). In contrast, others concentrated on training a generative model to synthesize new data that can substitute for the old data Shin et al. (2017); Van de Ven & Tolias (2018); Lavda et al. (2018); Ramapuram et al. (2020).

**Regularization-based Strategy.** This strategy prevents catastrophic forgetting by introducing a regularization term in the loss function to penalize the changes in the network parameters. Existing works can be divided into data-focused and prior-focused methods De Lange et al. (2021). The Data-focused methods take the previous model as the teacher and the current model as the student, transferring the knowledge from the teacher model to the student model through knowledge distillation. Typical methods include LwF Li & Hoiem (2017), LFL Jung et al. (2016), EBLL Rannen et al. (2017), DMC Zhang et al. (2020) and GD-WILD Lee et al. (2019). The prior-focused methods estimate a distribution over the model parameters, assigning an importance score to each parameter and penalizing the changes in significant parameters during learning. Relevant works include SI Zenke et al. (2017), EWC Kirkpatrick et al. (2017), RWalk Chaudhry et al. (2018a), AGS-CL Jung et al. (2020) and IMM Lee et al. (2017).

**Parameter Isolation-based Strategy.** This strategy considers dynamically modifying the network architecture by pruning, parameter mask, or expansion to greatly or even completely reduce catastrophic forgetting. Existing works can be roughly divided into two categories. One is dedicated to isolating separate sub-networks for each task from a large network through pruning techniques and parameter masks, including PackNet Mallya & Lazebnik (2018), PathNet Fernando et al. (2017), HAT Serra et al. (2018) and Piggyback Mallya et al. (2018). Another class of methods dynamically expands the network architecture, increasing the number of neurons or sub-network branches, to break the limits of expressive capacity (Rusu et al., 2016; Aljundi et al., 2017; Xu & Zhu, 2018; Rosenfeld & Tsotsos, 2018). However, as the number of tasks growing, this approach also complicates the network architecture and increases the computation and memory consumption.

**Gradient Orthogonal Projection-based Strategy.** Methods based on GOP strategies, which reduce catastrophic forgetting by projecting gradient or features with orthogonal projectors, have been shown to be effective in continual learning with encouraging results Farajtabar et al. (2020); Zeng et al. (2019); Saha et al. (2021); Wang et al. (2021); Chaudhry et al. (2020). According to the different ways of finding the projector, we can further divide the existing works into **C**ontext **O**rthogonal **P**rojection (COP) and **S**ubspace **O**rthogonal **P**rojection (SOP). Methods based on COP, such as OWM Zeng et al. (2019), Adam-NSCL Wang et al. (2021), and GPM Saha et al. (2021), always rely on the context of previous tasks to build projectors. In contrast to COP, SOP-based methods such as ORTHOG-SUBSPACE Chaudhry et al. (2020) use hand-crafted, task-specific orthogonal projectors and yield competitive results.

A related work to ours is TRGP Lin et al. (2022), which leverages the parameters of the most relevant old tasks for the new task to enhance forward knowledge propagation. The task-correlation

is computed by the norm of gradient projection onto the input subspace of each task. Unlike TRGP, LcSP does not depend on the Single Value Decomposition (SVD) algorithm to obtain the projector. Instead, LcSP derives the projector by minimizing the task coherence on the Oblique Manifold, where task coherence is the measure of alignment between projectors. Our experiments show that LcSP surpasses TRGP on such as Split CIFAR100 and miniImageNet benchmarks. Moreover, LcSP has low computational overhead and is faster than TRGP.

## 3    CONTINUAL LEARNING SETUP

In continual learning, the learner needs to learn multiple tasks sequentially. Let us assume that there are $T$ tasks, denoted by $\mathcal{T}_t$ for $t = 1, \ldots, T$ with its training data $\mathcal{D}_t = \{(x_i, y_i, \tau_t)_{i=1}^{N_t}\}$. Here, the data $(x_i, y_i) \in \mathcal{X} \times \mathcal{Y}_t$ is assumed to be drawn from some independently and identically distributed random variables, and $\tau_t \in \mathcal{T}$ denotes the task identifier. In the TIL setting, the data $\mathcal{D}_t$ can be accessed if and only if task $\mathcal{T}_t$ arrives. When episodic memory is adopted, a limited number of data samples drawn from old tasks can be stored in the replay buffer $\mathcal{M}$ so that $\mathcal{D}_t \cup \mathcal{M}$ can be used for training when task $\mathcal{T}_t$ arrives.

Assuming that a network $f$ parameterized with $\Phi = \{\theta, \varphi\}$ consists of two parts, where $\theta$ denotes the parameters of the backbone network and $\varphi$ denotes the parameters of the classifier. Let $f(x; \theta) : \mathcal{X} \times \mathcal{T} \to \mathcal{H}$ denote the backbone network parameterized with $\theta = \{W_l\}_{l=1}^L$, which encodes the data samples $x$ into feature vector. Let $f(x; \varphi) : \mathcal{H} \to \mathcal{Y}$ denote the classifier parameterized with $\varphi = w$ which returns the classification result of the feature vector obtained by $f(x; \theta)$. The goal of TIL is to learn $T$ tasks sequentially with the network $f$ and finally achieve the optimal loss on all tasks.

**Evaluation Metrics**    Once the training on all tasks is finished, we evaluate the performance of algorithm by calculating the average accuracy $\mathcal{A}$ and forgetting $\mathcal{F}$ Chaudhry et al. (2020) of the network on the T tasks $\{\mathcal{T}_1, ..., \mathcal{T}_T\}$. Suppose all tasks come sequentially, let $Acc_{i,j}$ denote the test accuracy of the network on task $\mathcal{T}_i$ after learning task $\mathcal{T}_j$, where $i \leq j$. The average accuracy is defined as

$$\mathcal{A} = \frac{1}{T} \sum_{i=1}^{T} Acc_{i,T}, \tag{1}$$

and the forgetting is defined as

$$\mathcal{F} = \frac{1}{T-1} \sum_{i=1}^{T-1} \max_{j \in \{i, ..., T-1\}} (Acc_{i,j} - Acc_{i,T}). \tag{2}$$

## 4    CONTINUAL LEARNING IN LOW-COHERENCE SUBSPACES

In this section, we describe the details of the LcSP algorithm based on hierarchical projection. In addition, LcSP can also be extended to more GOP methods. In the Appendix A.2, we provide the LcSP algorithm based on GPM, which can be used in class-incremental setting. In the following, we begin by introducing how to find task-specific, low-coherence projectors for LcSP on the Oblique manifold. We then describe how to use it in a specific DNN architecture to project features and gradients. Finally, we analyze the factors that enable LcSP to maintain plasticity and stability.

### 4.1    PRELIMINARY

Since our proposed algorithm involves knowledge related to optimization on oblique manifold, we first introduce the related mathematical definitions and concepts here to help readers better understand the context.

Optimization on the Oblique manifold, i.e., the solution lies on the Oblique manifold, is a well-established area of research Absil et al. (2009); Absil & Gallivan (2006); Selvan et al. (2012). Here, we briefly summarize the main steps of the optimization process. We refer readers who are interested in the relevant content to Absil et al. (2009) for more details.

Formally, the Oblique manifold $\mathcal{OM}(n, p)$ is defined as

$$\mathcal{OM}(n, p) \triangleq \{X \in \mathbb{R}^{n \times p} : \mathrm{diag}(X^\top X) = I_p\}, \tag{3}$$

representing the set of all $n \times p$ matrices with normalized columns. $\mathcal{OM}$ can also be considered as an embedded Riemannian manifold of $\mathbb{R}^{n \times p}$, endowed with the canonical inner product

$$\langle X_1, X_2 \rangle = \mathrm{trace}\left(X_1^\top X_2\right), \tag{4}$$

where $\mathrm{trace}(\cdot)$ represents the sum of the diagonal elements of the given matrix. For a given point $X$ on $\mathcal{OM}$, the tangent space at $X$, denoted by $T_X \mathcal{OM}$, is defined as

$$T_X \mathcal{OM}(n, p) = \{U \in \mathbb{R}^{n \times p} : \mathrm{diag}(X^\top U) = 0\}. \tag{5}$$

Further, the tangent space projector $\mathbf{P}_X$ at $X$ which projects $H \in \mathbb{R}^{n \times p}$ into $T_X \mathcal{OM}$, is represented as

$$\mathbf{P}_X(H) = H - X \, \mathrm{ddiag}\left(X^\top H\right), \tag{6}$$

where $\mathrm{ddiag}$ sets all off-diagonal entries of a matrix to zero. When optimizing on $\mathcal{OM}$, the $k$th iteration $X_k$ must move along a descent curve on $\mathcal{OM}$ for the cost function, such that the next iteration $X_{k+1}$ will be fixed on the manifold. This is achieved by the retraction

$$\mathbf{R}_{X_k}(U) = \mathrm{normalize}(X_k + U), \tag{7}$$

where $\mathrm{normalize}$ scales each column of the input matrix to have unit length. Finally, with this knowledge, we can extend the gradient descent algorithm to solve any unconstrained optimization problems on $\mathcal{OM}$, which can be summarized as

$$\begin{aligned} U &= \mathbf{P}_{X_k}(\nabla_{X_k} \mathbf{J}), \\ X_{k+1} &= \mathbf{R}_{X_k}(-\alpha U), \end{aligned} \tag{8}$$

where $\mathbf{J}$ denotes the cost function and $\nabla_{X_k} \mathbf{J}$ denotes the Euclidean gradient at the $k$th iteration and $\alpha$ is the step size.

## 4.2 Constructing Low-coherence Projectors on Oblique Manifold

In the following, we first introduce the concept of coherence metric. The coherence metric is usually used in compressed sensing and sparse signal recovery to describe the correlation of the columns of a measurement matrix Candes et al. (2011); Candes & Romberg (2007). Formally, the coherence of a matrix $M$ is defined as

$$\mu(M, N) = \begin{cases} \max_{j<k} \frac{|\langle M_j, M_k \rangle|}{\|M_j\|_2 \|M_k\|_2}, & M = N; \\ \max_{i,j} \frac{|\langle M_i, N_j \rangle|}{\|M_i\|_2 \|N_j\|_2}, & M \neq N, \end{cases} \tag{9}$$

where $M_j$ and $M_k$ denote the column vectors of matrix $M$. Without causing confusion, we use $\mu(M)$ to denote $\mu(M, M)$. To measure the coherence between different projectors, we introduce the Babel function Li & Lin (2018), measuring the maximum total coherence between a fixed atom and a collection of other atoms in a dictionary, which can be described as follows.

$$\mathbf{B}(P, M) = \max_{i \in M} \sum_{j \in P} \frac{|\langle M_i, P_j \rangle|}{\|M_i\| \|P_j\|} \tag{10}$$

where $M$ denotes a fixed atom and $P$ denotes target projector. With the concept of a coherence metric in mind, we then introduce the main optimization objective in finding projectors. Specifically, suppose that the DNN has learned the task $\mathcal{T}_1, \mathcal{T}_2, ..., \mathcal{T}_{t-1}$ in the subspace $\mathcal{S}_1, \mathcal{S}_2, ..., \mathcal{S}_{t-1}$, respectively, $P_1, P_2, .., P_{t-1}$ denote the projectors of all previous tasks. When learning task $\mathcal{T}_t$, we project features and gradients into a $d_t$-dimensional low-coherence subspace $\mathcal{S}_t$ with projector $P_t$ so that the LcSP can prevent catastrophic forgetting. The projector $P_t$ can be found by optimizing

$$\begin{aligned} &\arg\min \mathbf{B}(P_t, \mathbf{M}), \\ &\text{s.t.} \quad P_t \in \mathbb{R}^{m \times m}, \quad \mathrm{rank}(P_t) = d_t. \end{aligned} \tag{11}$$

Here $\mathbf{M} = \{P_1, ..., P_{t-1}\}$ denotes the collection of projectors of previous tasks.Two considerations need to be taken in solving Eq. (11), i.e., considering the rank constraint and the column vector's

scale (L2 norm) constraint. Empirically, extremely large or small length (L2 norm) of the projected column vector can lead to unstable training, as shown in Appendix A.1. We constrain the length of the projected column vectors to be equal to 1 because when the projection matrix $P_t$ satisfies the constraints and is orthogonal, the length of the gradient does not change after it has been projected, and thus does not affect the convergence rate. Therefore, we rephrase the rank and scale constrained problem as a problem on the Oblique manifold $\mathcal{OM}(m, d_t)$, by setting $P_t = O_t O_t^\top, O_t \in \mathbb{R}^{m \times d_t}$, and normalizing the columns of $O_t$, i.e, $\mathrm{diag}(O_t^\top O_t) = I_n$, where $\mathrm{diag}(\cdot)$ represents the diagonal matrix and $I_n$ is the $n \times n$ identity matrix.

With these settings, the new cost function $\mathbf{J}(\cdot)$ and the optimization problem can be described as follows:

$$\mathbf{J}(O_t, \mathbf{M}) = \begin{cases} \lambda \cdot \mathbf{B}(O_t O_t^\top, \mathbf{M}) + \gamma \cdot \mu(O_t O_t^\top), & t > 1 \\ \mu(O_t O_t^\top), & t = 1 \end{cases},$$

$$O_t = \arg\min \mathbf{J}(O_t, \mathbf{M}), \quad \text{s.t.} \quad O_t \in \mathcal{OB}(m, d_t). \tag{12}$$

In the cost function $\mathbf{J}(O_t, \mathbf{M})$, we define an inter-task optimization objective $\mathbf{B}(O_t O_t^\top, \mathbf{M})$, which measures the coherence between the current task projector $P_t$ and the previous task projectors $P_i$ $(i < t)$. Following the intuition of the GOP method, we minimize $\mathbf{B}(O_t O_t^\top, \mathbf{M})$ to reduce the interference between tasks, thereby overcoming catastrophic forgetting. In contrast, we define an intra-task optimization objective $\mu(O_t O_t^\top)$, which measures the coherence of the current task projector $P_t$ itself. We observe that using a projector with a particularly low rank means selecting a small portion of parameters to learn new tasks and will severely limit the model's plasticity, resulting in poor performance on new tasks. Therefore, we make $O_t$ full-rank by minimizing $\mu(O_t O_t^\top)$, in order to maintain the model's learning ability for new tasks. For the first task, since there is no interference from other tasks, we only need to focus on $\mu(O_t O_t^\top)$. When learning the $t$-th task $(t > 1)$, we consider both $\mathbf{B}(O_t O_t^\top, \mathbf{M})$ and $\mu(O_t O_t^\top)$, and to cope with different scenarios more flexibly, we utilize parameters $\gamma$ and $\lambda$ to provide a trade-off between them. In the Appendix A.1, we provide relevant ablation experiments and numerical analysis and summarize our algorithm for finding $O_t$ in $\mathcal{OM}$ for task $\mathcal{T}_t$.

## 4.3 APPLICATION OF LOW-COHERENCE PROJECTORS IN DNNS

With the LcSP at hand, the following introduces some technical details of applying LcSP in DNNs. When learning task $\mathcal{T}_t$, LcSP first constructs task-specific projector $P_t^l$ for each layer before training, and freezes them during training. These projectors are used to project the features and gradients, ensuring that the DNN learns in the low-coherence subspace. Specifically, suppose that a network $f$ with $L$ linear layers admits a DNN architecture, let $W_t^l$, $x_t^l$, $z_t^l$, $\sigma^l$, and $P_t^l$ denote the model parameters, the input features, the output features, the activation function, and the introduced low-coherence projector in layer $l \in \{1, ..., L\}$, respectively. LcSP introduces $P_t^l$ immediately after $W_t^l$ such that the pre-activation features are projected into the subspace, i.e.,

$$z_t^l = (x_t^l W_t^l) P_t^l,$$
$$x_t^{l+1} = \sigma^l(z_t^l). \tag{13}$$

According to the chain rule of derivation, the gradients at $W_t^l$ will also be multiplied with $P_l^t$ in backpropagation, as follows

$$\frac{\partial \mathcal{L}}{\partial (W_t^l)_{(i,:)}} = \frac{\partial \mathcal{L}}{\partial z_t^l} \frac{\partial z_t^l}{\partial (W_t^l)_{(i,:)}}$$

$$= \frac{\partial \mathcal{L}}{z_t^L} \prod_{k=l}^{L-1} \frac{\partial z_t^{k+1}}{\partial z_t^k} \cdot (x_t^l)_i \cdot P_t^l, \tag{14}$$

where $(W_t^l)_{(i,:)}$ represents the $i$th row of $W_t^l$ and $(x_t^l)_i$ is the $i$th element of $x_t^l$. In **C**onvolutional **N**eural **N**etworks (CNNs), the input and the output typically represent the image features and have more than two dimensions, e.g., input channel, output channel, height, and width. In this case, we reshape $z^l \in \mathbb{R}^{c_{\text{out}} \times (c_{in} \cdot h \cdot w)}$ to $z^l \in \mathbb{R}^{(c_{in} \cdot h \cdot w) \times c_{\text{out}}}$ and align the dimension of projector with the output channel so that $P_t^l \in \mathbb{R}^{c_{\text{out}} \times c_{\text{out}}}$. After the projection, we recover the shape of $z_t^l$ so that it can be used as input for the next layer.

### 4.4 METHOD ANALYSIS

In this section, we provide an analysis on the plasticity and the stability of LcSP.

**Stability Analysis.** Let $\theta = \{W_t^l\}_{l=1}^L$ denote the parameter set of $f$; $\Delta\theta = \{\Delta W_t^1, \ldots, \Delta W_t^L\}$ denote set of variation values of parameters after learning task $\mathcal{T}_t$; $P_t = \{P_t^l\}_{l=1}^L$ denote the projectors set obtained by LcSP; $x_{q,t}^l$ and $z_{q,t}^l$ denote the input and output when feeding the data of task $\mathcal{T}_q$ ($q \leq t$) into the network $f$, which has been optimized in learning task $\mathcal{T}_t$.

**Lemma 1.** *Assume that $f$ is fed the data of task $\mathcal{T}_t$ ($q < t$), then $f$ can effectively overcomes catastrophic forgetting if*

$$z_{q,q}^l \approx z_{q,t}^l, \quad \forall q \leq t \tag{15}$$

*holds for $l \in \{1, 2, ..., L\}$.*

Lemma 1 suggests that $f$ can overcome catastrophic forgetting if the output of $f$ to previous tasks is invariant. In the following, we prove that LcSP achieves approximate invariance to the output of previous tasks.

*Proof.* Suppose $q = t - 1$. When $l = 1$, $x_{q,t}^l = x_{q,q}^l$. Then

$$\begin{aligned}
z_{q,t}^l &= x_{q,t}^l(W_q^l + \Delta W_t^l)P_q^l \\
&= x_{q,t}^l W_q^l P_q^l + x_{q,t}^l \Delta W_t^l P_q^l \\
&= z_{q,q}^l + x_{q,t}^l \Delta W_t^l P_q^l.
\end{aligned} \tag{16}$$

Let $g_t^l$ denote the gradient when training the network on task $\mathcal{T}_t$. In backpropagation, $\Delta W_t^l = g_t^l P_t^l$. Then $x_{q,t}^l \Delta W_t^l P_q^l = x_{q,t}^l g_t^l P_t^l P_q^l$. If the inter-task coherence $\mu(P_t^l, P_q^l) \approx 0$, then $P_t^l P_q^l \approx \mathbf{0}$. Projectors satisfying this condition can be found by LcSP. We can prove that $z_{q,q}^l \approx z_{q,t}^l$ holds for all layers by repeating the above process. □

This proof can also be generalized to any previous task $\mathcal{T}_q$.

**Plasticity Analysis.** Let $\tilde{g}_t^l = g_t^l P_t^l$ denote the projected gradient at $W_t^l$. $f$ can achieve optimal loss on task $\mathcal{T}_t$ if $\langle g_t^l, \tilde{g}_t^l \rangle > 0$ holds for each $l \in \{1, \ldots, L\}$, where $\langle \cdot, \cdot \rangle$ represents the inner product. Here, we prove that $\langle g_t^l, \tilde{g}_t^l \rangle > 0$ holds for each $l \in \{1, \ldots, L\}$.

**Proof.** *Let $\tilde{g}_t^l = g_t^l P_t^l$ denote the projected gradient, we have*

$$\begin{aligned}
\langle g_t^l, \tilde{g}_t^l \rangle = g_t \tilde{g}_t^{l\top} &= g_t^l O_t^l O_t^{l\top} g_t^{l\top} \\
&= \langle g_t^l O_t^l, g_t^l O_t^l \rangle = \|g_t^l O_t^l\| > 0.
\end{aligned} \tag{17}$$

*Note that $\|g_t^l O_t^l\|$ is always positive unless $g_t^l O_t^l$ is $\mathbf{0}$. This result is easy to generalize to each layer.*

## 5 EXPERIMENTS

In this section, we evaluate our approach on several popular continual learning benchmarks and compare LcSP with previous state-of-the-art methods. The result of accuracy and forgetting demonstrate the effectiveness of our LcSP, especially when the number of tasks is large.

### 5.1 BENCHMARKS

We evaluate the effectiveness of our algorithms in several widely used continuous learning datasets: Permuted MNIST, Rotated MNIST, Split CIFAR100, and Split miniImageNet. The Permuted MNIST dataset is derived from MNIST LeCun (1998) by randomly permuting the image pixels with different seeds for different tasks. The Rotated MNIST dataset is another variation of MNIST that rotates the images by a random angle between $[0, \pi]$ for each task. For both Permuted MNIST and Rotated MNIST, we generate 10 sequential tasks with 10 classes each. The Split CIFAR100 dataset is obtained by dividing CIFAR100 into 20 tasks, where each task contains five randomly selected classes (without replacement) from the total of 100 classes. The Split miniImageNet dataset, used in Chaudhry et al.

Table 1: The average accuracy and forgetting results of the proposed LcSP and baselines.

| Methods | Permuted MNIST | | Rotated MNIST | | Split CIFAR100 | | Split miniImageNet | |
|---|---|---|---|---|---|---|---|---|
| | $\mathcal{A}(\%)$ | $\mathcal{F}$ | $\mathcal{A}(\%)$ | $\mathcal{F}$ | $\mathcal{A}(\%)$ | $\mathcal{F}$ | $\mathcal{A}(\%)$ | $\mathcal{F}$ |
| EWC Kirkpatrick et al. (2017) | 89.97 | 0.04 | 92.68 | 0.03 | 68.80 | 0.02 | 52.01 | 0.12 |
| A-GEM Chaudhry et al. (2018b) | 83.56 | 0.14 | 93.36 | 0.02 | 63.98 | 0.15 | 57.24 | 0.12 |
| ER-Res Chaudhry et al. (2019) | 87.24 | 0.11 | 94.16 | 0.01 | 71.73 | 0.06 | 58.94 | 0.07 |
| HAT Serra et al. (2018) | - | - | - | - | 72.06 | 0.00 | 59.78 | 0.03 |
| OWM Zeng et al. (2019) | 90.71 | 0.01 | 93.35 | 0.01 | 50.94 | 0.30 | - | - |
| GPM Saha et al. (2021) | 93.91 | 0.03 | 95.22 | 0.01 | 72.48 | 0.00 | 60.41 | 0.00 |
| Adam-NSCL Wang et al. (2021) | - | - | - | - | 75.95 | 0.04 | 63.27 | 0.06 |
| TRGP Saha et al. (2021) | **96.34** | 0.01 | **96.79** | 0.01 | 74.46 | 0.01 | 61.78 | 0.01 |
| ORTHOG-SUBSPACE Chaudhry et al. (2020) | 93.43 | 0.02 | 94.46 | 0.01 | 64.30 | 0.07 | 51.40 | 0.10 |
| **LcSP (ours)** | 95.16 | 0.02 | 96.12 | 0.01 | **76.47** | 0.00 | **67.90** | 0.00 |

Table 2: Total training time measured on a single GPU after learning all the tasks. The training time is normalized with respect to the value of GPM. We refer Saha et al. (2021) for a specific value.

| Methods | Training Time [s] | | |
|---|---|---|---|
| | Permuted MNIST | Split CIFAR100 | Split miniImageNet |
| EWC Kirkpatrick et al. (2017) | 2.63 | 1.76 | 1.22 |
| A-GEM Chaudhry et al. (2018b) | 1.82 | 3.48 | 2.19 |
| ER-Res Chaudhry et al. (2019) | 1.06 | 1.49 | **0.82** |
| HAT Serra et al. (2018) | - | 1.62 | 0.90 |
| OWM Zeng et al. (2019) | 6.77 | 2.41 | - |
| Adam-NSCL Wang et al. (2021) | - | 1.20 | 1.51 |
| ORTHOG-SUBSPACE Chaudhry et al. (2020) | 1.72 | 1.90 | 3.69 |
| TRGP Saha et al. (2021) | 3.03 | 3.36 | 4.39 |
| GPM Saha et al. (2021) | 1.00 | 1.00 | 1.00 |
| **LcSP (ours)** | **0.90** | **0.71** | 0.95 |

(2018b), is created by splitting 100 classes of miniImageNet into 20 sequential tasks with 5 classes each. In addition, we conducted fair comparison experiments using the same settings on the longer Permuted MNIST (containing 150 tasks) and Permuted CIFAR10 (containing 64 tasks). Compared to baselines, we verified that LcSP is better able to maintain the learning capability on long task sequences.

## 5.2 BASELINES

We compare the proposed method with several state-of-the-art approaches that consider sequential task learning in fixed network architecture. These approaches include GOP-based methods, such as Orthogonal Weight Modulation (OWM) Zeng et al. (2019), Adam-NSCL Wang et al. (2021), Gradient Projection Memory (GPM) Saha et al. (2021), Trust Region Gradient Projection (TRGP) Lin et al. (2022) and ORTHOG-SUBSPACE Chaudhry et al. (2020), regularization-based methods, such as HAT Serra et al. (2018) and Elastic Weight Consolidation (EWC) Kirkpatrick et al. (2017), and replay-based methods, such as Experience Replay with reservoir sampling (ER-Res) Chaudhry et al. (2019) and Averaged GEM (A-GEM) Chaudhry et al. (2018b).

## 5.3 IMPLEMENTATION DETAILS

For experiments on Permuted MNIST, we use a fully connected network with two hidden layers, each of which is with 256 neurons, by utilizing ReLU activations. Consistent with GPM, we use a 5-layer AlexNet Krizhevsky et al. (2012) for experiments on CIFAR100 and a standard ResNet18 for experiments on miniImageNet. For experiments on MNIST, all tasks share the same classifier. For experiments on CIFAR and miniImageNet, each task requires a task-specific classifier. For all experiments, LcSP does not use episodic memory to store data samples for data replay. For all methods, we use **S**tochastic **G**radient **D**escent (SGD) uniformly. The learning rate is set to 0.01 for experiments on MNIST and 0.003 for experiments on CIFAR and ImageNet. Both $\lambda$ and $\gamma$ in Eq. (12) are set to 1. All experiments were run five times with five different random seeds.

## 5.4 MAIN RESULTS

**Permuted MNIST and Rotated MNIST.** In this experimental setup, a single-head classifier is employed for all tasks. HAT and Adam-NSCL are excluded from the comparison as require a

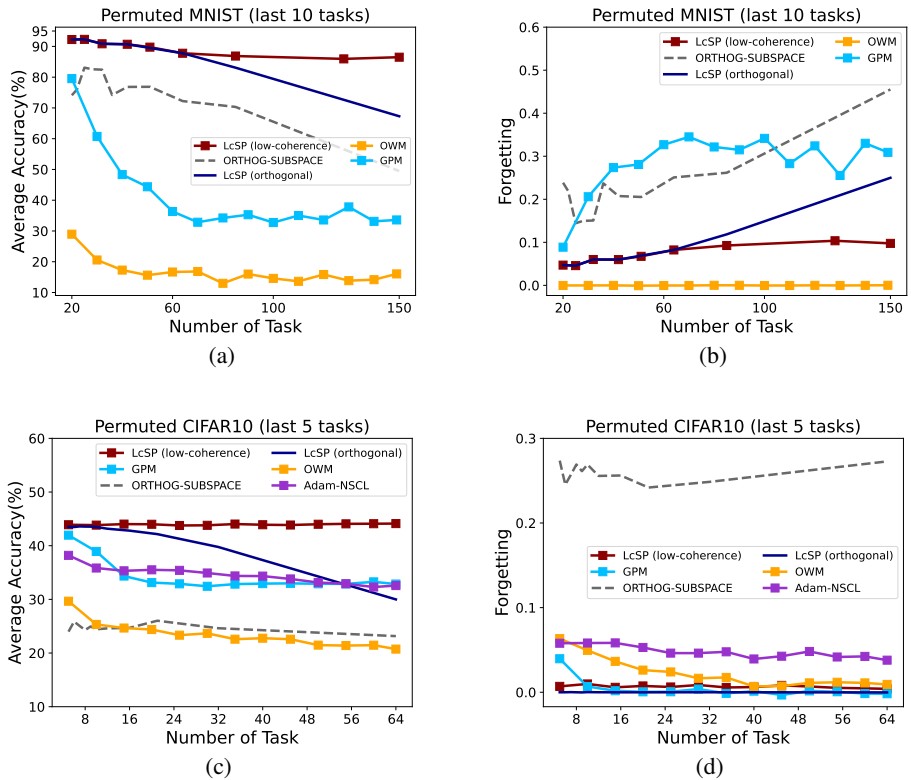

Figure 1: (a) and (b) show the average accuracy and forgetting of the last 10 tasks on Permuted MNIST when learning 150 tasks. (c) and (d) show the average accuracy and forgetting of the last 5 tasks on Permuted CIFAR10 when learning 64 tasks.

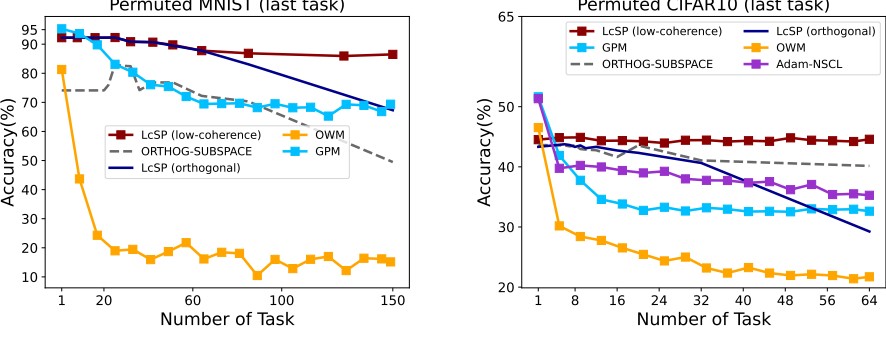

Figure 2: The accuracy of the last task on Permuted MNIST (left) and Permuted CIFAR10 (right), respectively.

separate classifier for each task. Tab. 3 presents that LcSP obtained competitive results on MNIST. LcSP outperformed other baselines while was slightly inferior to TRGP, with average accuracies of 95.16% and 96.12%, respectively. We found that LcSP outperformed other methods mainly due to its hierarchical projection mechanism, which effectively minimized task interference. Moreover, by keeping the projectors with low coherence, LcSP could utilize the network capacity more efficiently. However, LcSP also had some limitations. It did not apply projection to the classification layer, which resulted in more severe forgetting on this layer compared to TRGP. Although LcSP was exceeded by TRGP in average accuracy, it had a considerably shorter training time than TRGP, as shown in Tab. 2, which made LcSP more efficient and practical in comparison.

**Split CIFAR100.** In this experiment, we adopted the multi-head setup, which enabled us to compare with HAT and Adam-NSCL. As shown in Tab. 3, LcSP outperformed all baselines, achieving an

average accuracy of 76.47%, exceeding the baselines by $23.53\% \sim 0.52\%$, and marginally surpassing Adam-NSCL. Moreover, our results showed that LcSP achieved zero forgetting. This was explained by two factors. First, by using the distinct classification heads for each task, LcSP avoided forgetting on the classification layer. Second, due to the ample network capacity to accommodate all tasks, LcSP did not need to compromise some stability for plasticity.

**Split miniImageNet.** In this experiment, we assessed the effectiveness of our algorithm on a deeper network (standard ResNet18). Tab. 3 shows that our method achieved a remarkable improvement in average accuracy over the baseline methods, from $15.89\%$ to $4.53\%$. This indicates that LcSP has good scalability on deep neural networks and can be applied to large datasets and more complex tasks.

**Comparisons of Learning** 150 **Tasks and** 64 **Tasks.** To demonstrate the promising advantage of the proposed methods in learning a long sequence of tasks, the following experiments compare the results with 64 tasks and 150 tasks. Note that, in Fig. 1, LcSP (orthogonal) is a variant that uses orthogonal projectors, while LcSP (low-coherence) uses low-coherence projectors. Figs. 1(a) and 1(b) report the average accuracy and forgetting of the last 10 tasks, with learning 150 tasks on Permuted MNIST. Figs. 1(c) and 1(d) report the average accuracy and forgetting of the last 5 tasks with learning 64 tasks on Permuted CIFAR10. The average accuracy of all methods, except LcSP (low-coherence), dramatically declines or remains low as the number of tasks increases. Furthermore, it can be seen from Fig. 1(d) that all methods except ORTHOG-SUBSPACE have almost no forgetting. This result indicates that methods using orthogonal projectors gradually lose their learning capacity with the increasing number of tasks. The proposed method uses the low-coherence projector to relax the orthogonal constraint, effectively solving this problem.

**Efficiency Analysis.** To evaluate the practicality of the LcSP, we measured the total training time of all algorithms on a single GPU, normalized by the time of GPM. As shown in Tab. 2, the proposed LcSP trains faster than all baselines on MNIST and Split CIFAR100, and slightly slower than HAT and ER-Res on Split miniImageNet. The main reasons why LcSP training is faster than other GOP-based baselines are as follows. Firstly, LcSP uses dimensionally aligned projectors to project features, which is faster than manual projection of gradients (e.g. Adam-NSCL, GPM, and TRGP) during backpropagation. Secondly, LcSP trains the network parameters directly in Euclidean space, whereas ORTHOG-SUBSPACE trains the network parameters on Stiefel manifolds. This makes LcSP faster than ORTHOG-SUBSPACE, especially on deep neural networks. Thirdly, in contrast to TRGP which uses the post-projection gradient length to calculate task correlation and trust region projections, LcSP only needs to calculate the coherence of the projectors to evaluate task coherence. Since the number of parameters of the projectors is much smaller than the number of parameters of the model, LcSP can be trained faster than TRGP.

## 5.5 ABLATION STUDIES

**Learning Capacity Degradation in Gradient Orthogonal Projection.** To further investigate the learning capacity degradation problem, we give the accuracy of baselines for the last task on Permuted MNIST and Permuted CIFAR10. As shown in Fig. 2, all baselines, except LcSP, suffer from this problem with different degrees and show some decrease in accuracy compared to the initial ($66.16\% \sim 24.63\%$ on Permuted MNIST and $24.8\% \sim 3.48\%$ on Permuted CIFAR10). These results suggest that learning capacity degradation is the critical factor that results in degrading the performance of GOP-based methods in the case of a large number of tasks.

## 6 CONCLUSION

This paper experimentally observes that GOP methods in continual learning suffer from the learning capacity degradation problem. Specifically, the performance of the GOP methods on new tasks gradually decreases as the number of tasks increases. This paper proposed a novel method, namely LcSP, to address this problem. Instead of learning in orthogonal subspace, LcSP projects features and gradients via low-coherence projectors to minimize inter-task and intra-task coherence. Extensive experiments show that our approach works well in alleviating forgetting and has a significant advantage in maintaining learning capacity, especially in learning long-sequence tasks. In future work, the LcSP can be extended to more continual learning methods, and improve the learning capability of DNN models with larger number of tasks.

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

# A APPENDIX

In this section, we provide supplementary ablation experiments and numerical analysis to support our main findings. Moreover, we also present the comparison results of LcSP and the baseline methods in terms of inference speed. Finally, we summarize the two algorithmic procedures of LcSP, which are based on hierarchical projection and GPM respectively.

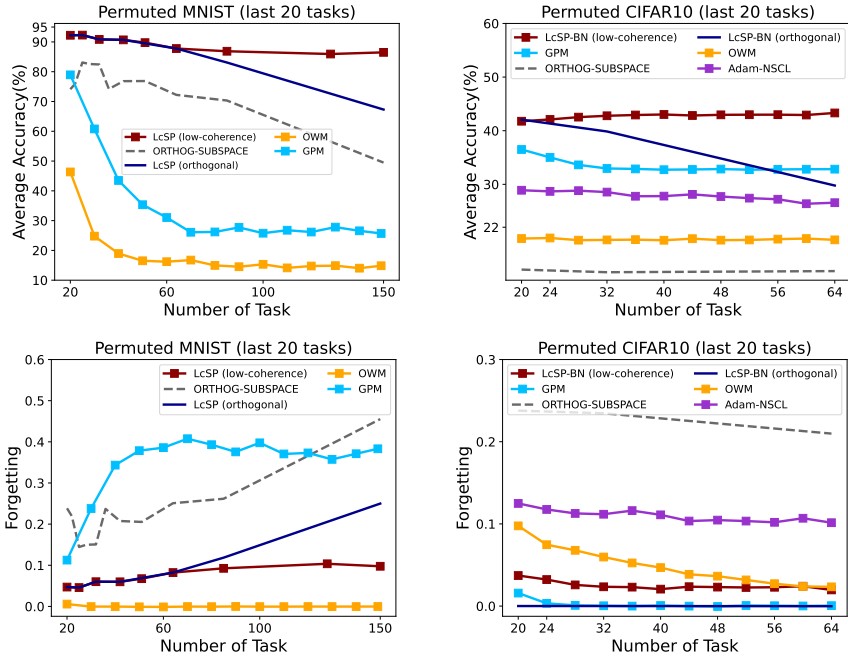

Figure 3: The average accuracy and forgetting of the last 20 tasks on Permuted MNIST (left) and Permuted CIFAR10 (right). The x-axis shows the number of tasks learned and y-axis represents the corresponding average accuracy and forgetting on the last 20 tasks.

## A.1 ABLATION EXPERIMENTS AND NUMERICAL ANALYSIS

In Tab. 3, we provide the average accuracy and forgetting rate with the standard deviation values on different datasets. Tab. 4 demonstrates the computational overhead of the low-coherence projector construction algorithm. In Tab. 5 and Tab. 6, we provide a comparison of computational overhead for forward reasoning between LcSP and the baseline approaches. In Tab. 7 and Tab. 8, we provide preliminary experimental results of LcSP on ImageNet-1k.

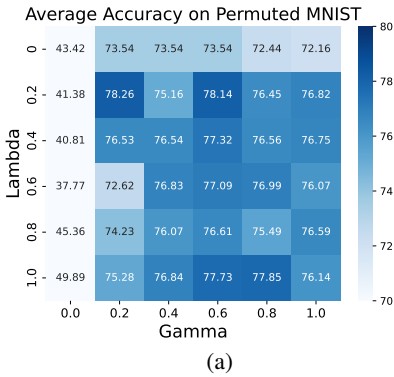 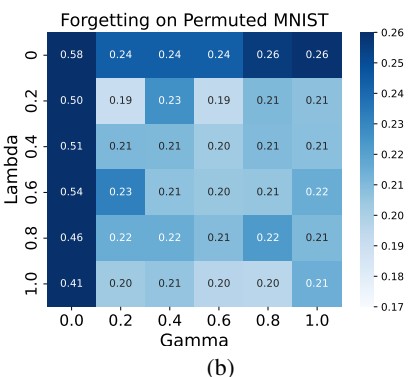

(a)                (b)

Figure 4: Average accuracy and forgetting for different $\lambda$ and $\gamma$ on Permuted MNIST. A fully connected network with 2 hidden layers, each with 64 neurons, is used for this experiment.

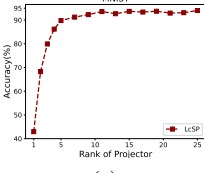 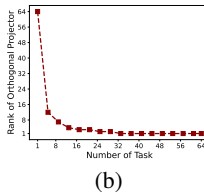 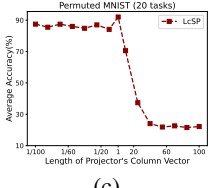 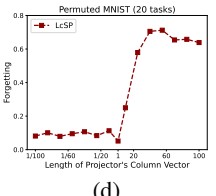

(a)         (b)         (c)         (d)

Figure 5: (a) gives the ablation study for different ranks of projectors on MNIST. (b) shows the average rank of all projectors when the number of tasks increases. Here, the dimension $m$ of features is $64$. (c) and (d) show accuracy and forgetting performance when columns of projectors have different scales.

**Additional results on Permuted MNIST and Permuted CIFAR10**    Readers may wonder whether our conclusion holds if we evaluate the average performance with more tasks (e.g., the average accuracy and forgetting on the last 20 tasks). As shown in Fig.3, LcSP still outperforms all baselines with a significant advantage. However, the phenomenon of learning capacity degradation in baselines becomes more imperceptible, e.g., the average accuracy of OWM on Permuted CIFAR10 is consistently low, rather than significantly decreasing. Fig. 4 gives the ablation study and shows the performance of our method with different $\lambda$ and $\gamma$. When $\lambda$ equals $\gamma$, the result of average accuracy on Permuted MNIST reached the highest. Results reached the worst when either $\lambda$ or $\gamma$ is equal to zero. These results indicate that both inter-task and intra-task coherence should be minimized to solve the plasticity and stability dilemma.

**Ablation Studies and Experiments for Rank and Scale Constraints.** Further ablation studies and experiments are conducted to investigate the effects of the rank and scale constraints on the expressive power (plasticity) and stability of DNNs. The result in Fig. 5(a) suggests that projecting features or gradients into subspaces with low dimensions (e.g., lower than $5$ in Fig. 5(a) ) will decrease the expressive power of a DNN. Finally, Fig. 5(c) and Fig. 5(d) give an ablation study for scale constraints on the projector's columns. In Fig. 5(c) and Fig. 5(d), when the columns of the projector have unit length, the average accuracy reaches the highest. The result gets worse when the length of the projector's columns is too small or too large.

## A.2    ALGORITHM PSEUDO CODE

In the following, we provide pseudo-code implementations of two algorithms, i.e., the Low Coherence Projector Construction Algorithm (shown in Algorithm 1) and the Continual Learning Algorithm with GPM based on LcSP (shown in Algorithm 2).

Table 3: The average accuracy and forgetting results along with the standard deviation values on different datasets. **Memory** denotes whether the method is trained using a replay strategy with episodic memory.

| Methods | Memory | Permuted MNIST (10 tasks) | | Permuted MNIST (20 tasks) | |
|---|---|---|---|---|---|
| | | Accuracy(%) | Forgetting | Accuracy(%) | Forgetting |
| EWC | ✗ | 89.97(±0.57) | 0.04(±0.01) | 92.68(±0.76) | 0.03(±0.01) |
| AGEM | ✓ | 83.56(±0.16) | 0.14(±0.01) | 93.36(±0.42) | 0.02(±0.01) |
| ER-Res | ✓ | 87.24(±0.31) | 0.11(±0.01) | 94.16(±0.31) | 0.01(±0.01) |
| OWM | ✗ | 90.71(±0.11) | 0.01(±0.00) | 93.35(±0.79) | 0.01(±0.00) |
| GPM | ✗ | 93.91(±0.16) | 0.03(±0.00) | 95.22(±0.23) | 0.01(±0.02) |
| TRGP | ✗ | **96.34**(±**0.11**) | 0.01(±0.00) | **96.59** (±**0.23**) | 0.01(±0.02) |
| ORTHOG-SUBSPACE | ✗ | 93.43(±0.03) | 0.02(±0.01) | 94.46(±0.91) | 0.01(±0.01) |
| **LcSP** | ✗ | 95.16(±0.46) | 0.02(±0.01) | 96.12(±0.21) | 0.01(±0.01) |

| Methods | Memory | Split CIFAR100 | | Split miniImageNet | |
|---|---|---|---|---|---|
| | | Accuracy(%) | Forgetting | Accuracy(%) | Forgetting |
| EWC | ✗ | 68.80(±0.88) | 0.02(±0.01) | 52.01(±2.53) | 0.12(±0.03) |
| AGEM | ✓ | 63.98(±1.22) | 0.15(±0.02) | 57.24(±0.72) | 0.12(±0.01) |
| ER-Res | ✓ | 71.73(±1.19) | 0.06(±0.01) | 58.94(±2.92) | 0.07(±0.01) |
| HAT | ✗ | 72.06(±0.50) | 0.00(±0.00) | 59.78(±0.57) | 0.03(±0.00) |
| OWM | ✗ | 50.94(±0.60) | 0.30(±0.01) | - | - |
| ORTHOG-SUBSPACE | ✓ | 64.3(±0.59) | 0.07(±0.01) | 51.4(±1.44) | 0.10(±0.01) |
| GPM | ✗ | 72.48(±0.40) | 0.00(±0.00) | 60.41(±0.61) | 0.00(±0.00) |
| Adam-NSCL | ✗ | 75.95 | 0.04 | 63.27 | 0.06 |
| TRGP | ✗ | 74.46(±0.32) | 0.01(±0.00) | 61.78(±0.60) | 0.01(±0.01) |
| **LcSP (ours)** | ✗ | **76.47**(±**0.36**) | 0.00(±0.00) | **67.9**(±**0.40**) | 0.00(±0.00) |

Table 4: Convergence Speed and Time Cost of Constructing $P_t$ by Optimizing Loss Function $J(O_t)$ in ResNet18. The **stage** Represents the block structure in ResNet18.

| | **stage1** | **stage2** | **stage3** | **stage4** |
|---|---|---|---|---|
| dims | 64 | 128 | 256 | 512 |
| times (seconds) | 0.01 | 0.02 | 0.10 | 1.04 |
| iterations | 15 | 15 | 19 | 27 |
| cost val | $3.01 \times 10^{-16}$ | $5.69 \times 10^{-14}$ | $5.97 \times 10^{-14}$ | $9.28 \times 10^{-16}$ |
| grad L2 norm | $6.93 \times 10^{-8}$ | $9.54 \times 10^{-7}$ | $9.78 \times 10^{-7}$ | $1.22 \times 10^{-7}$ |

Table 5: Comparison of Inference Computational Cost Based on AlexNet.

| | OWM | GPM | TRGP | LcSP |
|---|---|---|---|---|
| FLOPs | 26,676,256 | 23,202,880 | 23,284,800 | 27,523,680 |
| Mean Inference Time (ms) | 1.249 | 1.152 | 1.270 | 1.474 |

Table 6: Comparison of Inference Computational Cost Based on ResNet18.

| | Adam-NSCL | ORTHOG-SUBSPACE | TRGP | LcSP |
|---|---|---|---|---|
| FLOPs | 558,538,572 | 556,708,864 | 558,548,992 | 558,965,376 |
| Mean Inference Time (ms) | 10.089 | 3.957 | 5.335 | 5.893 |

Table 7: Performance of LcSP based on ResNet18 on ImageNet-1k: We focus on the average accuracy and forgetting rate of the last 5 tasks of LcSP to validate whether the model can maintain learning ability while overcoming catastrophic forgetting. There are a total of 25 tasks, each with 40 classes.

| | 5 | 10 | 15 | 20 | 25 |
|---|---|---|---|---|---|
| Average Accuracy (latest 5 tasks) | 52.53 | 51.68 | 52.31 | 51.26 | 51.72 |
| Forgetting | 0.04 | 0.05 | 0.04 | 0.04 | 0.05 |

Table 8: Performance of LcSP based on ResNet18 on ImageNet-1k: Here we focus on the average accuracy and forgetting rate of the last 20 tasks of LcSP. There are a total of 100 tasks, each with 10 classes.

|  | 20 | 40 | 60 | 80 | 100 |
|---|---|---|---|---|---|
| Average Accuracy (latest 20 tasks) | 47.19 | 46.52 | 47.38 | 48.26 | 48.31 |
| Forgetting | 0.09 | 0.09 | 0.08 | 0.08 | 0.09 |

---

**Algorithm 1** Construct the $O_t$ on $\mathcal{OM}$ for Task $\mathcal{T}_t$

---

1: **function** FIND_PROJECTOR($\mathcal{M}$) :
2: **Input:** $\mathcal{M} \leftarrow \{P_1, \ldots, P_{t-1}\}$
3: **Output:** $O_t$
4: $\mathbf{R}_X(U) := \text{normalize}(X + U)$
5: $X_0 \leftarrow$ random initialization on $\mathcal{OM}$
6: $k \leftarrow 0$
7: Set tolerance error $0 \leq \mathcal{E} \ll 1$
8: **while** True **do**
9:    $G \leftarrow \nabla f(X_k)$
10:   $U \leftarrow G - X_k \, \text{ddiag}(X_k^\top G)$
11:   **if** $\|U\| \leq \mathcal{E}$ **then**
12:      break
13:   **end if**
14:   $\alpha \in (0, 0.5), \quad \beta \in (0, 1)$
15:   $t \leftarrow 1$
16:   **while** $\mathbf{J}(\mathbf{R}_{X_k}(-t \cdot U), \mathcal{M}) > \mathbf{J}(X_k, \mathcal{M}) - \alpha \cdot t \cdot \|U\|_2^2$ **do**
17:      $t \leftarrow \beta \cdot t$
18:   **end while**
19:   $X_{k+1} \leftarrow \mathbf{R}_{X_k}(-t \cdot U)$
20:   $k \leftarrow k + 1$
21: **end while**
22: $O_t = X_k$
23: **Return** $O_t$

---

**Algorithm 2** Algorithm for Continual Learning with GPM based on LcSP

---

1: $\mathbf{M}^l \leftarrow []$, for all $l = 1, 2, ..., L$
2: $\mathcal{M} \leftarrow \{(\mathbf{M})_L^l\}$
3: $\mathbf{W} \leftarrow \mathbf{W}_0$
4: **for** $\tau \in 1, 2, ..., T$ **do**
5:   $P_\tau \leftarrow$ FIND_PROJECTOR($\mathcal{M}$)
6:   **repeat**
7:     $B_n \sim \mathcal{D}_\tau^{train}$
8:     $\nabla_\mathbf{W} L_\tau \leftarrow$ SGD($B_n, f_\mathbf{W}$)
9:     $\nabla_\mathbf{W} L_\tau \leftarrow$ PROJECTION($\nabla_\mathbf{W} L_\tau, P_\tau$)
10:     $\mathbf{W} \leftarrow \mathbf{W} - \alpha \nabla_\mathbf{W} L_\tau$
11:   **until** convergence
12:   $B_{n_s} \sim \mathcal{D}_\tau^{train}$
13:   $\mathbf{R}_\tau \leftarrow$ forward($B_{n_s}, f_\mathbf{W}$), where $\mathbf{R}_\tau = \{(\mathbf{R}_\tau^l)_{l=1}^L\}$
14:   **for** layer$, l = 1, 2, ..., L$ **do**
15:     $\hat{\mathbf{R}}_\tau^l \leftarrow$ PROJECTION($\mathbf{R}_\tau^l, \mathbf{M}^l$)
16:     $\hat{\mathbf{U}}_\tau^l \leftarrow$ SVD($\mathbf{R}_\tau^l$)
17:     $k \leftarrow$ criteria($\hat{\mathbf{R}}_\tau^l, \mathbf{R}_\tau^l, \epsilon_{th}^l$)
18:     $\mathbf{M}^l \leftarrow \left[\mathbf{M}^l, \hat{\mathbf{U}}_\tau^l[0:k]\right]$
19:   **end for**
20: **end for**
21: **Return** $f_\mathbf{W}, \mathcal{M}$

---

