# OpenReview forum: "Low-coherence Subspace Projection: Enhance the Learning Capacity of Orthogonal Projection Methods on Long Task Sequences"
_ICLR.cc/2024/Conference — Submitted to ICLR 2024_

### Official Review · Reviewer_g3YD · 2023-10-19

**Soundness:** 3 good
**Presentation:** 3 good
**Contribution:** 2 fair
**Rating:** 5
**Confidence:** 4

**Summary:**

This paper finds that GPM maintains stability when learning new tasks. However, the flexibility of learning new tasks in a long task sequence is constrained due to the orthogonal projection mechanism of GPM. To address this issue, this paper proposes to optimize on oblique manifold to relax the orthogonal projection constraint. Specifically, the proposed method minimizes the cost function that involves both inter-task and intra-task coherence. The paper also provides some theoretical analysis to support the idea. Extensive experiments on several standard CL benchmarks show the improvement of the LcSP compared to GPM and other related methods.

**Strengths:**

* The motivation of the proposed method is clear.


* This paper optimizes on oblique manifold to relax the orthogonal constraint in GPM to improve the flexibility of learning new tasks.

**Weaknesses:**

* The idea of using an oblique manifold to relax the orthogonal constraint in GPM is not new. For example, [1] employs similar ideas to improve the flexibility of learning new tasks in GPM.  They also relax the orthogonal constraint in GPM. Further, another alternative solution is to assign different weights to the bases so that new tasks can learn more effectively, as proposed by [2].  These similar works diminish the novelty of this work.

* Optimization in oblique manifold is a well-studied technique in existing literature. Thus, the technical contribution of this paper is weak.

* The authors state that their method can be applied to class-incremental learning. The reviewer did not find any experiment evaluations about class-incremental learning setting.

Reference:

[1]   Restricted Orthogonal Gradient Projection for Continual Learning. 2023

[2]   Continual Learning with Scaled Gradient Projection. AAAI 2023

**Questions:**

N/A

---

### Official Review · Reviewer_UiYh · 2023-10-27

**Soundness:** 3 good
**Presentation:** 1 poor
**Contribution:** 3 good
**Rating:** 5
**Confidence:** 2

**Summary:**

This paper proposes a new method to mitigate catastrophic forgetting for continual learning.
Instead of orthogonal subspace, which is used in previous methods, this paper uses low-coherence subspaces
by optimizing parameter projectors to minimize the coherence of matrices and Babel function with constraints on the oblique manifold.
The effectiveness of the proposed method is evaluated on continuous learning datasets:
Permuted MNIST, Rotated MNIST, Split CIFAR100, and Split miniImageNet.

**Strengths:**

- The motivation of this paper is good, and the approach seems reasonable. This paper criticizes the limited representation power of orthogonal projection methods, and low-coherence projection seems a reasonable relaxation of orthogonal projections to address the problem.
- Experimental results seem to show that the proposed method outperforms the baselines. The accuracy of TRGP is slightly larger than the proposed method, but the proposed method is more efficient than TRGP. Figs. 1 and 2 show the superiority of the proposed method clearly.
- This paper attempts to guarantee the performance theoretically. However, as in the following form, theoretical results are not very clearly written.

**Weaknesses:**

This paper is not well written. I think that this paper presents an interesting idea, but careful writing of the article is a minimum requirement for acceptance so that readers do not misunderstand.
- Writings of theoretical results are a bit odd. Lemma 1 seems to define the condition (eq. (15)) when models overcome catastrophic forgetting, but in afterward proof, the paper proves that the proposed method satisfies the condition. I think that definition is separated from Lemma 1 and Lemma 1 should be finished like that "models trained by LcSP satisfy eq. (15)".
The second proof does not have a theoretical claim, and it is difficult to grasp what result is proved by this proof.
Additionally, I suggest you to add intuitive explanation of theoretical results after each claim.
Please note that I am not an expert in this area. So, plasticity and stability might be well known in this area, and these results may not require additional explanations for experts.
Even so, I think this paper should show the definition of stability and plasticity or citation that presents these concepts before Method Analysis.
- This paper may have been written in haste and contains many typographical errors. For examples, Singular value decomposition is written as "Single" Value Decomposition.
This paper uses wrong table number in Section of Experiments: Table 3 is referred in the main paper but it is the table in appendix.
Many careless errors undermine the reliability of this paper: It is difficult to believe that an experiment is being performed without error or bugs when there are so many errors in writing of the paper.
- Several papers in Reference lack the title of journal or conference.

**Questions:**

I might have misunderstood the paper.  If so, please point out such parts in my review and correct them.

---

### Official Review · Reviewer_6aA9 · 2023-11-04

**Soundness:** 2 fair
**Presentation:** 2 fair
**Contribution:** 2 fair
**Rating:** 3
**Confidence:** 4

**Summary:**

Applying regularization (such as orthogonal regularization) to the network representation subspace is a common practice in continual learning to handle the forgetting issue. This paper proposes to apply the regularization in low-coherence subspace by learning on the Oblique manifold. Task-specific projections are learned to encourage that the representations corresponding to different tasks are in low-coherence subspaces.

**Strengths:**

- The paper is well-written, though there are a few areas that could benefit from improvement, as discussed below.
- This paper uses the regularization in the low-coherence subspace rather than the orthogonal subspace, enabling the model to utilize its capacity more effectively, resulting in reduced parameter interference and mitigated capacity degradation. This proposition is well-founded.
- The experiments show the advantages of learning in the low-coherence subspace in comparison to the orthogonal subspace.

**Weaknesses:**

- It has been extensively studied to use orthogonal subspace-based regularization to alleviate forgetting in continual learning. The main idea of this paper is to replace the orthogonal subspace modeling on Riemannian manifold with the low-coherence subspace modeling on Oblique manifold. The contribution and novelty are somehow incremental, especially considering the issues mentioned below.

- It is clear that the orthogonal subspace learning can alleviate the forgetting by “isolating” the tasks in the representation subspace. As a relaxation of the orthogonal subspace, it is unclear how well the low-coherence subspace-based method can perform on handling the forgetting issue, in priciples. The limits and assumptions behind the proposed method are not clearly analyzed and discussed.

- The developed method (with task-specific projection) and experiments are limited to only the task-incremental continental learning, where the task identifiers are required in both training and testing. Although the authors mention an algorithm to apply the low-coherence subspace modeling with other methods to handle other CL settings, they seem not implemented and validated in the experiments. And the settings of experiments are not introduced clearly. It influences the significance of the proposed method.

**Questions:**

- Please address the questions in "Weaknesses".
- Why the reported experimental results of the compared methods (such as ORTHOG-SUBSPACE Chaudhry et al. (2020)) are different from their papers? It seems the network backbone, datasets, and settings are the same.

---

### Meta-Review · Area_Chair_9uEK · 2023-12-04

**Metareview:**

The reviewers were unanimous in their vote to reject, feeling that the submission was not ready for publication. There was no author response to the issues raised by the reviewers.

**Justification For Why Not Higher Score:**

Reviewers do not support acceptance.

**Justification For Why Not Lower Score:**

N/A

---

### Decision · Program_Chairs · 2024-01-16

Reject